# Effects of Water Exposure on the Interfacial Bond between an Epoxy Resin Coating and a Concrete Substrate

**DOI:** 10.3390/ma12223715

**Published:** 2019-11-11

**Authors:** Boreum Won, Min Ook Kim, Sangmin Park, Jin-Hak Yi

**Affiliations:** 1School of Ocean Science and Technology, Korea Maritime and Ocean University, Busan 49112, Korea; brwon@kiost.ac.kr; 2Coastal Development and Ocean Energy Research Center, Korea Institute of Ocean Science and Technology, Busan 49111, Korea; sangmin@kiost.ac.kr (S.P.); yijh@kiost.ac.kr (J.-H.Y.)

**Keywords:** underwater application, epoxy resin coating, concrete substrate, interfacial bond, coating equipment, submerged structures

## Abstract

Interfacial bond properties of six different epoxy resins used to coat submerged concrete structures were investigated. Test variables included coating type, coating equipment, and underwater curing time. Coating thickness and pull-off bond strength were measured using commercially available test equipment. Coating thickness and bond strength varied greatly depending on the manufacturer. The standard (control) coating equipment positively influenced the bond strength compared to other equipment. The effect of curing time on the bond properties was not significant within the range of 24 to 72 h. Lastly, some important considerations for the underwater coating of actual marine and coastal concrete structures were discussed, and suggestions for future research are presented.

## 1. Introduction

Marine and coastal concrete structures are directly exposed to harsh environments, which can lead to rebar corrosion owing to chloride ion penetration into the concrete. Surface protection prior to immersion in seawater is particularly important to prevent such penetration [1,2,3,4,5,6,7,8,9,10,11,12,13,14,15,16,17,18]. Underwater coating technology has been widely accepted for repair work of marine structures, including submerged concrete columns, dock walls, and floating concrete structures. It has many advantages over other types of remediation, such as procedural simplicity, length of time investment, and affordability [19]. The coating material type, coating equipment, local environmental conditions, and condition of the damaged structure are all important considerations for successful underwater coating. However, the selection of appropriate coating material and coating equipment is based on the experience of individual applicators owing to the lack of practical guidelines and standard testing procedures.

Many experimental studies have been reported as providing useful information regarding the performance evaluation and/or selection of coating materials for concrete application [20,21,22,23,24,25,26,27]. Medeiros and Helene [20] found that polyurethane is more effective than acrylic at ensuring concrete durability, based on Rapid Chloride Permeability Test (RCPT) test results. Merah et al. [21] confirmed that epoxy resin coatings show a higher carbonation resistance than that of acrylic coatings. Li et al. [22] investigated the time-dependent changes in carbonation resistance of three different coatings (polyurethane, epoxy, and chlorinated rubber) applied to a concrete surface and confirmed that polyurethane performed best under the same conditions. Vipulanandan and Liu [23] focused on the long-term performance of polyurethane-coated concrete located in corrosive environments and observed that bond strength can change significantly depending on the surface condition of the concrete substrate prior to coating. Moradllo et al. [24] conducted five-year exposure tests to evaluate the durability performance of coated concrete with six different coatings and reported that polyurethane and acrylic coatings were the two most efficient coatings at reducing chloride ion penetration into concrete. These authors also noted that the performance of a surface coating can be degraded over time and that re-coating must be carried out to ensure the design service life of concrete structures. Lie and Vipulanandan [25] reported that measured bond strength values can vary with conditions in the testing environment and recommended following procedures such American Society for Testing and Materials (ASTM) C321 and ASTM D4541. Al-Kheetan et al. [26] investigated the effect of early-age water exposure on both the bond property and water absorption of two different cementitious coatings (crystalline and polymer). Coated concrete specimens were placed in a water tank after curing in air for 24 h and samples were removed for testing after 24, 72, 120, 240, 308, and 408 h. The results showed that bonding of the polymeric cementitious coating was better than that of the crystalline cementitious coating, and the effect of surface roughness on bond strength was negligible, as shown in Figure 1. Kim et al. [27] recently reported that the tensile bond strength of epoxy coatings applied to wet concrete is generally lower compared with that applied to dry concrete. They concluded that the outside air temperature is one of the most important factors that must be considered prior to underwater coating owing to its significant influence on pot life.

According to previous studies, epoxy resin is one of most commonly utilized coatings for the surface protection of concrete structures. Such studies primarily relied on accelerated chloride penetration tests and/or bond strength measurements to evaluate performance. Some researchers observed that performance can vary between manufacturers, even for generic types of coating. It should be noted that most previous studies were conducted in air, rather than underwater; thus, the effect of water exposure on bond performance was not systematically investigated. Therefore, the objective of this study was to identify the effects of several important parameters on the performance of underwater coatings, specifically, the type of coating, coating equipment, and curing time. To this end, six commercially available underwater coating materials were carefully selected, and bond performance was evaluated using coating thickness and pull-off bond strength values. The underwater coating was conducted in a water-filled cylindrical tank. Three types of coating equipment were used to evaluate the effect of application method on bond strength and coating thickness.

## 2. Research Significance

Although underwater coating has been widely utilized in repairing submerged structures, the age-dependency of interfacial adhesion between applied coating and substrates has not been systematically investigated. Based on the premise that bond properties of underwater coatings change over time and long-term performance of marine structures is linked to coating bonding, it is critical for an ocean engineer to select the optimal coating material and application method. Understanding age-dependent change in tensile bond characteristics can provide engineers with guidelines for determining the optimal coating material and coating method. Thus, a comprehensive experimental program was carried out to investigate the effects of different coatings and coating equipment on the age-dependent bond properties of underwater coatings. Coating thickness and pull-off bond strength between the applied epoxy resin coating and concrete substrate were measured and the results were compared between the different treatments.

## 3. Experimental Investigation

### 3.1. Materials and Test Variables

Six different epoxy resin coatings, three types of coating equipment, and three curing periods were considered as test variables in this study. The six underwater coatings were labelled C1 (Alocit 28.14, A&E Group, Shah Alam, Malaysia), C2 (Alocit 28.15, A&E Group, Shah Alam, Malaysia), C3 (Belzona 5831, Belzona, Harrogate, England, UK), C4 (RS 500P, Chemco, Coatbridge, Scotland, UK), C5 (RA 500UW, Chemco, Coatbridge, Scotland, UK), and C6 (Intergard 5027, AkzoNobel, Amsterdam, Netherlands). All coating materials consisted of an epoxy resin and hardener. Material properties are summarized in Table 1. C1 and C2 are usually applied to steel and concrete underwater, respectively, while both C3 and C4 are used only for steel under conventional atmospheric conditions. Specifically, six equivalent resins from four different suppliers were considered in this study.

As shown in Figure 2, a conventional roller, electric airless sprayer, and portable gun-type applicator were selected as the coating equipment and labelled as control, E1, and E2, respectively. A conventional roller was defined as the control since it is one of the most common tools for underwater coating work. E1 is a commercially available electric airless texture sprayer (GRACO Mark V, GRACO Inc., Minneapolis, MN, US) that is generally used for the surface protection of ships and containers. E1 operates by sucking the coating liquid through its pump system and discharging it to a spray gun. E2 is a portable gun-type applicator based on a piston mechanism that was designed as part of this study. E2 consisted of a wool pad to apply the coating, a reservoir for the coating material, a coil spring, and a piston cylinder. The coating material was pumped to the wool pad from the reservoir by operating the piston cylinder. The coatings were applied underwater, then test pieces were immediately placed in a curing tank filled with water. It should also be noted that all the samples were prepared in underwater conditions.

Coated concrete specimens were cured for 24, 48, and 72 h, and measurements were conducted to study the early-age bond characteristics of selected coatings. Total curing time is the period between immersion of the coated specimen in the water-filled curing tank and the removal for testing. The effect of the concrete surface roughness on the bond was not included in the range of this study based on the results of previous studies [26,27].

### 3.2. Substrate Concrete and Composite Sample Fabrication

Table 2 shows the mix proportion of the concrete substrate. Test pieces were rectangular with dimensions of 300 × 300 × 30 mm. They were cast with a cement content of 340 kg/m^3^, a coarse/fine aggregate ratio of 1.39, and a water/cement ratio (*w*/*c*) of 0.24 to produce high-strength concrete. All test pieces were steam cured under controlled laboratory conditions. Compressive strength and splitting tensile strength were determined using three standard cylinders with a 100 mm diameter and 200 mm length and tested at 28 days. The average compressive strength and tensile strength of the substrate concrete were 82.3 MPa and 5.1 MPa, respectively.

The composite samples were fabricated and cured as follows:
Substrate concrete test pieces were cast as depicted in Figure 3a.At 24 h after casting, the test pieces were demolded and placed in a steam curing chamber.Within two days of casting, the concrete surface was roughened with high pressure water jetting, as shown in Figure 3b, and placed back in the chamber.Epoxy resin and a hardener were mixed for 2 to 5 min, which was well within the pot life of all the resins.Each coating was applied to the surface of concrete specimens in a cylindrical tank filled with tap water.Coated concrete specimens were then immediately moved to a curing tank filled with tap water and cured for 24, 48, or 72 h.After each curing time, specimens were taken out of the water tank for measurement of their coating thickness and pull-off bond strength.

The sample fabrication process is shown in Figure 4. Cleaning of coating equipment was conducted immediately after coating was completed. E1 and E2 were cleaned thoroughly using a solvent, such as a thinner or methyl ethyl ketone, and then the equipment was dried in air. A new roller was used for each coating application.

### 3.3. Test and Measurement

Coating thickness was measured using a concrete coating thickness gauge (Elcometer A500C-B, Elcometer, Manchester, England, UK, measurable thickness ranges from 0.15 to 2.50 mm) following the procedure of ASTM D6132 [28]. Prior to contacting a transducer to the specimen, an ultrasonic couplant gel was applied to the targeted location for effective ultrasonic wave propagation. Figure 5 shows the process of calibrating the ultrasonic gauge prior to measuring the coating thickness. Calibration foils of known thickness were used to obtain speed data. The average thickness was estimated from four points near the targeted location for bond strength measurements (Figure 4f). To evaluate the bond strength of each coating, the pull-off tension bond tests were performed in accordance with ASTM C1583 [29]. Equation (1) was used to calculate the pull-off bond strength:(1)$$f_{bond}=\frac{P_{failure}}{A}$$

Pull-off bond strength, *f_bond_*, was calculated by dividing the load at failure, *P_failure_*, by the bonded area, *A* (1962.5 mm^2^), formed using a glued-on dolly, where *P_failure_* is the peak load. A commercially available pull-off adhesion tester (Elcometer F510-50S, Elcometer, Manchester, England, UK, measurable bond strength ranges from 0.3 MPa to 4.0 MPa, pull-off rates: 0.02 MPa/sec) was used. The targeted locations for bond strength measurements were carefully selected and the dollies were glued using epoxy, as depicted in Figure 4g.

After curing the dolly adhesive for 24 h under constant conditions, the dollies were pulled off and the peak load was recorded. The effects of test variables on the interfacial bond between each epoxy coating and concrete substrate were analyzed based on the measured thickness and bond strength values. The test results were also compared with previous studies [26,27] to investigate the interfacial bond properties between the coating and concrete substrate under water exposure conditions.

## 4. Test Results and Discussion

### 4.1. Effects of Coating Material on Measured Thickness and Bond Strength

Figure 6 illustrates the effect of each coating on the measured coating thickness after curing underwater for 72 h. Thickness values varied greatly with the type of coating equipment utilized. Average thickness values were 0.95 mm, 0.36 mm, and 0.46 mm for the control, E1, and E2, respectively. The influence of coating materials on thickness was less obvious. The control C6 coating was thicker than those of C1–C5.

Figure 7 shows the effect of the different epoxy resins on the interfacial bond after 72 h of curing underwater. Average bond strength values (MPa) are recorded on the graphs. Bond strength varied depending on the type of coating equipment, with the highest values mainly attributed to the control. In general, C1 and C2 coatings exhibited the highest bond strengths. The C6 coating exhibited the worst bond performance, although bond strength improved when applied using the E1 and E2 equipment. The C1 and C2 bond strengths obtained in a previous study were 32% and 33% lower, respectively, than that obtained in this study [27]. This change in performance may be related to differences in the coating device and environmental conditions, such as temperature and humidity.

The effects of the selected epoxy resin coating on the measured thickness and bond strength values were not significant. It should be noted that this study focused on the effect of water exposure on the interfacial bond between the concrete substrate and the applied epoxy coating. Applied coatings can degrade during water exposure and human errors can influence the bond performance in the case of actual coating underwater. Therefore, additional research is important to investigate the effect of long-term exposure on bond properties. For a precise comparison of each resin, the coating weight must be constant.

### 4.2. Effects of Coating Equipment on Measured Coating Thickness and Bond Strength

Figure 8 shows the effect of equipment type on measured coating thickness and bond strength values. The thickness values for the E1 and E2 equipment were about 60% of the control equipment. Thickness values for E2 ranged between 0.20 mm and 0.46 mm, while other equipment showed a higher variation; i.e., E2 produced a thin and uniform coating compared to the other application methods. The average bond strength of coatings applied using the control method was 2.3 and 1.8 times higher than that of E1 and E2 equipment, respectively. Thus, it can be concluded that the control equipment contributed to the bond strength to a much greater extent the other two methods. In summary, the E2 equipment was found to be most advantageous in terms of a thin and uniform coating, while the control equipment showed the best performance in increasing the bond property.

### 4.3. Effects of Water Exposure Time on the Measured Coating Thickness and Bond Strength

Figure 9 and Figure 10 depict the effects of the water exposure time on the coating thickness and bond strength, respectively. Based on previous research [26], it was expected that the bond strength of the coating would increase over time; however, the effect of the exposure time on the bond strength was not clear. For example, even after curing for 72 h, some bond strength values were lower than after 24 h. In a previous study [26], coatings were applied to a concrete substrate under constant laboratory conditions and tested via exposure to water. The bond strengths of each coating increased with curing time. This difference might be related to the initial environmental conditions for the coatings rather than the curing time itself, i.e., the underwater coatings adopted in this study may be susceptible to washing off the substrate into the water. It can be concluded that 72 h was not a sufficient length of time to clarify the effects of water exposure on the coating thickness and bond strength. In this regard, further experimental studies are necessary to verify the effect of water exposure on coating bond properties.

### 4.4. Relationship between Thickness and Bond Strength

Figure 11 shows the relationship between the measured thickness and the bond strength values. Bond strength values varied between 0.1 MPa and 4.0 MPa when the coating thickness was less than 0.8 mm. Bond strength tended to decrease with increasing coating thickness. It was originally expected that as the coating thickness increased, adhesion would improve as well. Coating thickness may not have contributed to the increase of bond strengths because the coatings were not completely cured after 72 h underwater. As the coatings were not fully hardened, bond strength (adhesion strength) measurements were affected.

### 4.5. Discussion

The bond strength values measured throughout this study ranged between 0.30 MPa and 3.66 MPa, and showed good agreement with previous studies [26,27]. However, the measured values were generally lower (about 15% to 53%) than specified on the datasheets provided by several manufacturers (see Table 1). This can be related to the type of substrate material, i.e., the bond strengths provided by the manufacturers were measured using steel plates. Furthermore, bond strength values may vary greatly depending on environmental conditions, such as high temperature and humidity. According to a previous study [27], bond strengths may decrease by up to 39% when the coating is applied underwater. Thus, it can be concluded that the bond strength values measured in this study were lower than manufacturers’ specifications due to the coatings being cured underwater instead of under typical atmospheric conditions. In previous studies [26,27], only one type of coating equipment was utilized since the application method was not considered as a variable. In this study, it was confirmed that the coating equipment had a significant influence on both the early-age bond property and uniform thickness of coatings. The results indicate that the bond performance of the selected coatings varied by manufacturer, although the coating resins were of the same generic type. Therefore, a preliminary investigation must be conducted to determine the optimal coating equipment and highest-performing coating material for effective repair work of submerged concrete structures.

The bond properties of coatings may vary greatly depending on environmental conditions (temperature, humidity, and salinity); therefore, it is necessary to investigate bonding with results that are applicable to the marine environment. Table 3 shows the chemical composition of seawater that may affect the coating performance.

Most studies, including this investigation, have used tap water rather than seawater. Hence, chemical and physical deterioration factors in real conditions were not considered. In a marine environment, structures are continuously subjected to climatic factors (such as temperature, wind, and waves) and chemical deterioration factors (such as chloride and sulfate attacks). Temperature is an important factor that must be considered during coating due to its influence on coating pot life. Therefore, experimental studies using sea water that consider all relevant external factors are required. Further investigations are needed to provide more practical information on selecting an appropriate coating material and coating equipment for the underwater repair of submerged concrete structures.

## 5. Conclusions

This study aimed to investigate the effects of early water exposure on the bond properties of epoxy coatings used for the underwater repair of submerged concrete structures. Six epoxy resin coatings, three different coating tools, and three different underwater curing times were selected as test variables and their effects on interfacial bond between coating and substrate concrete were experimentally investigated. Based on the test results and comparisons, the following conclusions can be drawn:The effects of coating type on the measured thickness and bond strength values were not explicit and the bond strength values varied depending on the manufacturer.The effects of the selected coating equipment on the measured thickness and bond strength values were clear. The paint roller yielded a better bond performance than other tools.The effect of underwater curing time on the interfacial bond between the applied coating and substrate was not clearly observable within 72 h; hence, longer exposure periods are needed for future research.The bond strength values ranged between 0.1 MPa and 4.0 MPa when coating thickness values were less than 0.8 mm, and the bond strength tended to decrease with increasing coating thickness.

In this study, the bond performance of epoxy resin coatings that are commercially available was investigated by using three different coating tools in underwater conditions. Further experimental research that considers the marine environment are required to provide more useful information on the bond characteristics of underwater coatings and the durability of coated concrete in seawater.

## Figures and Tables

**Figure 1 materials-12-03715-f001:**
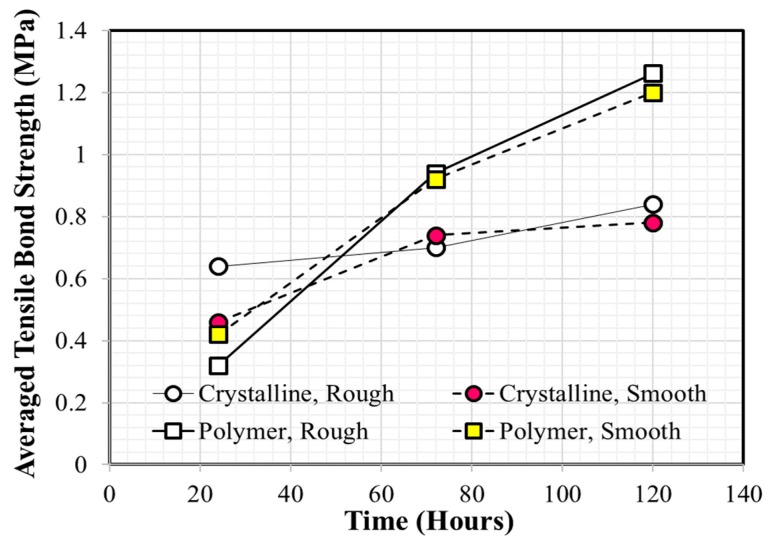
Average tensile bond strength values of crystalline and polymeric coatings (Redrawn using the data of Al-Kheetan et al. [26]).

**Figure 2 materials-12-03715-f002:**
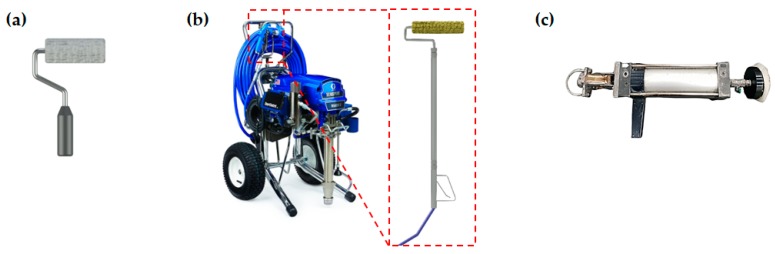
Selected coating equipment: (**a**) control, (**b**) E1, and (**c**) E2.

**Figure 3 materials-12-03715-f003:**
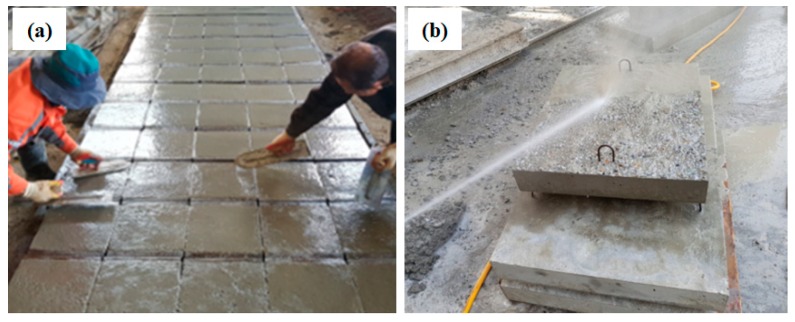
Concrete specimen casting and surface preparation procedure: (**a**) concrete substrate fabrication, and (**b**) water jetting to make a rough surface.

**Figure 4 materials-12-03715-f004:**
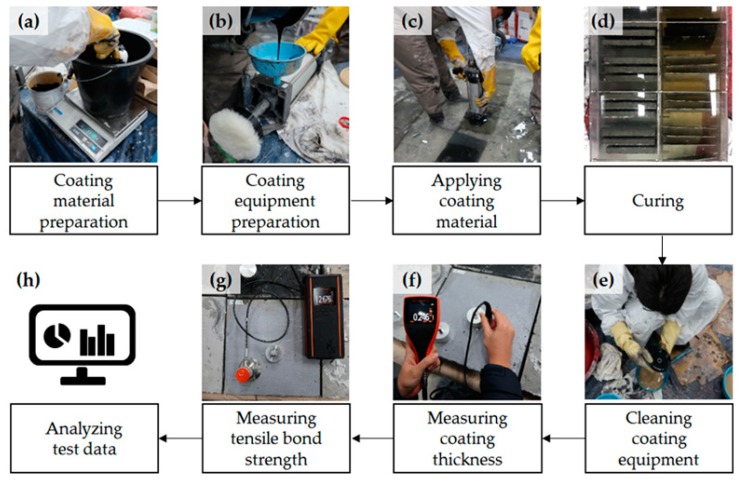
Coating, curing, and measurement process: (**a**) coating material preparation; (**b**) coating equipment preparation; (**c**) applying coating material; (**d**) curing; (**e**) cleaning; (**f**) coating thickness measurement; (**g**) bond strength measurement; (**h**) analyzing.

**Figure 5 materials-12-03715-f005:**
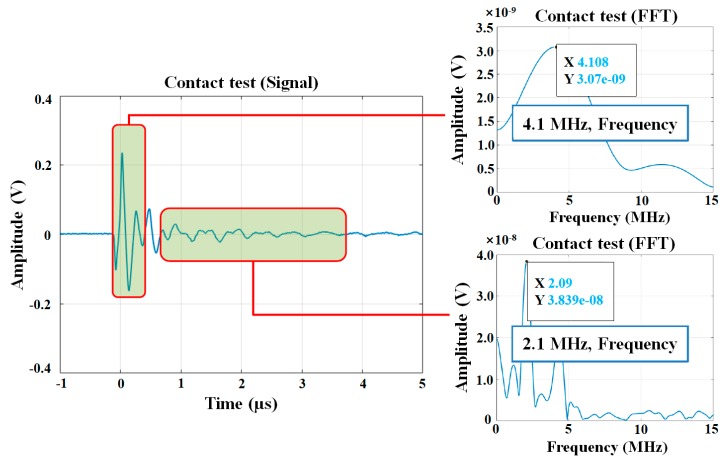
Calibration of the ultrasonic device for coating thickness measurement.

**Figure 6 materials-12-03715-f006:**
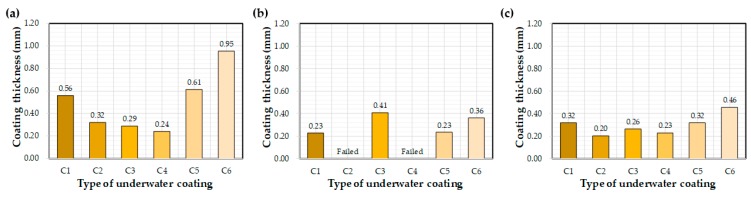
Average coating thickness after 72 h of water exposure: (**a**) control, (**b**) E1, and (**c**) E2.

**Figure 7 materials-12-03715-f007:**
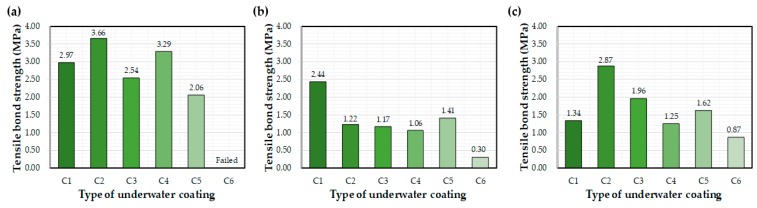
Average bond strength values after 72 h of water exposure: (**a**) control, (**b**) E1, and (**c**) E2.

**Figure 8 materials-12-03715-f008:**
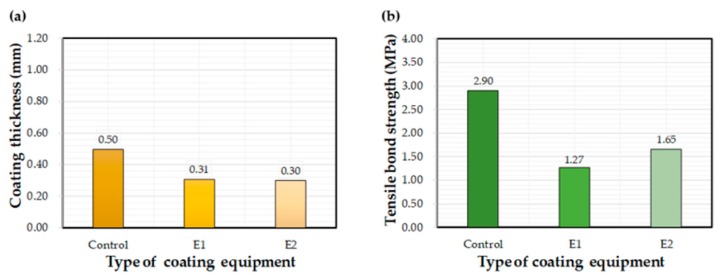
Effects of the selected coating equipment on (**a**) coating thickness and (**b**) bond strength.

**Figure 9 materials-12-03715-f009:**
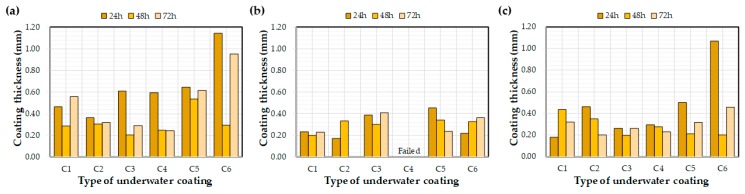
Effect of the water exposure time on the measured coating thickness: (**a**) control, (**b**) E1, and (**c**) E2.

**Figure 10 materials-12-03715-f010:**
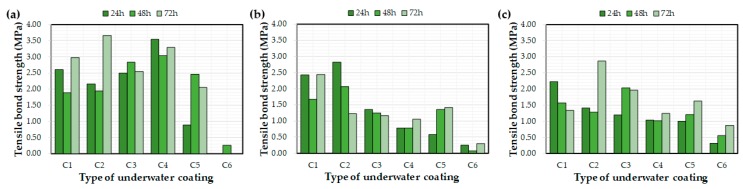
Effect of the water exposure time on the measured bond strength: (**a**) control, (**b**) E1, and (**c**) E2.

**Figure 11 materials-12-03715-f011:**
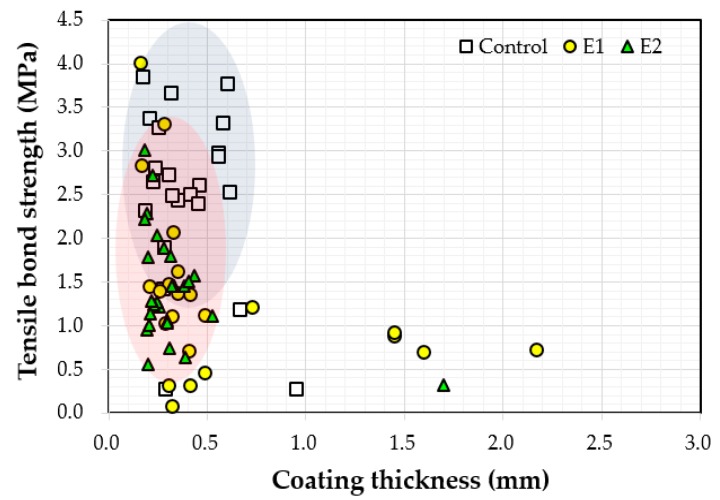
Correlation between the measured coating thickness and bond strength.

**Table 1 materials-12-03715-t001:** Material properties of the selected commercial epoxy resin coatings.

Epoxy Resin Coating (Code)	Mix Ratio (Resin:Hardener)	Density (g/cm^3^)	Bond Strength (MPa) (Measured Using Steel Plate)
C1	5:1	1.82	17.0
C2	5:1	1.55	6.9
C3	1.75:1	1.75	12.7
C4	5.1:1	1.60	16.6
C5	2.77:1	1.21	N/A
C6	2:1	N/A	N/A

**Table 2 materials-12-03715-t002:** Mix proportion and measured strength values for the substrate concrete.

Concrete Mix Proportion					Compressive Strength	Splitting Tensile Strength
Water (kg/m^3^)	Cement (kg/m^3^)	Fine Aggregate (kg/m^3^)	Coarse Aggregate (kg/m^3^)	Water Reducer (kg/m^3^)	*f’_c_* (MPa)	*f’_sp_* (MPa)
83	340	325	451	5.0	82.3	5.1

**Table 3 materials-12-03715-t003:** Chemical composition of seawater compared to tap water [30].

Element	Unit	Seawater	Tap Water
Calcium (Ca)	ppm	389	90
Chloride (Cl)		18,759	44
Iron (Fe)		0.5121	-
Potassium (K)		329	6
Magnesium (Mg)		1323	6
Sodium (Na)		9585	26
Sulfate (SO_4_^2−^)		831	8
Nitrate (NO_3_)		0.1345	1
Salinity	ppt	35	less than 0.5

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
