# Peer review of "Effects of Water Exposure on the Interfacial Bond between an Epoxy Resin Coating and a Concrete Substrate"

_materials, 2019, doi:10.3390/ma12223715_

Round 1

Reviewer 1 Report

Dear Autors,

I enjoyed reading this paper and found the results interesting. The introduction is clear and frames the need for this research well. The literature review is clear. The methods used in the paper are experimental methods. The methods are appropriate for the work and clearly described. The analysis of the results is sufficient. The authors report the outcomes and provide analysis of the results. A discussion section is good. The article has the correct structure. English is generally correct, moderate English changes required. Reference to figures and tables has been done correctly. The figures and tables are generally clear, except for one paragraph 119 is "dimensions 300 x 300 x 30 mm^3" should be "dimensions 300 x 300 x 30 mm". To conclude: the article is suitable for publication in its current form after minor corrections.

Author Response

Dear Reviewer,

I appreciate for your time and consideration for reviewing our manuscript.

To reflect received comments, the manuscript was re-reviewed by a native English speaker and editorial changes were made.

Reviewer 2 Report

There are no comments. A good, interesting and practically important manuscript.

Author Response

Dear Reviewer,

We appreciate for your time and consideration for reviewing our manuscript.

Reviewer 3 Report

The topic of the paper is interesting, especially for hydraulic and coastal engineers.

The scientific approach is sound, based on experimental analysis characterized by an innovative approach: the bond performance of coatings have been tested underwater. 

Author Response

(The authors gave the same response as above.)
